# From plans to actions in patient and public involvement: qualitative study of documented plans and the accounts of researchers and patients sampled from a cohort of clinical trials

Deborah Buck,[1] Carrol Gamble,[1] Louise Dudley,[1] Jennifer Preston,[2] Bec Hanley,[3] Paula R Williamson,[1] Bridget Young,[4] The EPIC Patient Advisory Group

For numbered affiliations see end of article.

**Correspondence to**
Professor Bridget Young;
byoung@liv.ac.uk

## ABSTRACT

Patient and public involvement (PPI) in research is increasingly required, although evidence to inform its implementation is limited.

**Objective:** Inform the evidence base by describing how plans for PPI were implemented within clinical trials and identifying the challenges and lessons learnt by research teams.

**Methods:** We compared PPI plans extracted from clinical trial grant applications (funded by the National Institute for Health Research Health Technology Assessment Programme between 2006 and 2010) with researchers' and PPI contributors' interview accounts of PPI implementation. Analysis of PPI plans and transcribed qualitative interviews drew on the Framework technique.

**Results:** Of 28 trials, 25 documented plans for PPI in funding applications and half described implementing PPI before applying for funding. Plans varied from minimal to extensive, although almost all anticipated multiple modes of PPI. Interview accounts indicated that PPI plans had been fully implemented in 20/25 trials and even expanded in some. Nevertheless, some researchers described PPI within their trials as tokenistic. Researchers and contributors noted that late or minimal PPI engagement diminished its value. Both groups perceived uncertainty about roles in relation to PPI, and noted contributors' lack of confidence and difficulties attending meetings. PPI contributors experienced problems in interacting with researchers and understanding technical language. Researchers reported difficulties finding 'the right' PPI contributors, and advised caution when involving investigators' current patients.

**Conclusions:** Engaging PPI contributors early and ensuring ongoing clarity about their activities, roles and goals, is crucial to PPI's success. Funders, reviewers and regulators should recognise the value of preapplication PPI and allocate further resources to it. They should also consider whether PPI plans in grant applications match a trial's distinct needs. Monitoring and reporting PPI before, during and after trials will help the research community to optimise PPI, although the need for ongoing flexibility in implementing PPI should also be recognised.

## Strengths and limitations of this study

- This was the first study to examine whether plans for patient and public involvement (PPI), as documented in trialists' grant applications, were subsequently implemented.
- Semistructured interviews with chief investigators and patients allowed us to identify challenges to implementing PPI, and lessons learnt, from a range of informant perspectives.
- The study benefited from the inclusion of a combination of trials which had ended at the time of the interviews, and those which were ongoing.
- Some informants struggled to recall events pertaining to PPI for trials which had ended—a drawback of retrospective study designs.
- We used a historical cohort of trials, funded 4–8 years previously. The emphasis on PPI has grown over these years, thus our findings may not fully reflect the planning and implementation of PPI in trials funded recently.

## INTRODUCTION

There are several schools of thought regarding why patient contributors should be involved as advisors or partners in healthcare research, rather than just as participants. Ethical and political arguments for patient partnerships are based on values such as democracy, accountability and empowerment.[1–3] Alongside these values are pragmatic arguments which revolve around the belief that patient and public involvement (PPI) can enhance the relevance, validity, quality and success of research.[1–5] The growth in PPI nationally and internationally[6–8] is reflected by its increasing assimilation into grant applications, with funding bodies encouraging researchers to submit plans for PPI in order to obtain funding.[2 9–12] Such developments have branched out into other realms

including patient involvement in academic publishing, for instance within *The BMJ*.[13]

For PPI contributors, getting involved in research has been reported to lead to 'personal development' such as boosting confidence, empowerment and a sense of purpose.[14] Similarly there can be personal benefits for researchers who have reported that their attitudes, values and beliefs about the worth of PPI had been heightened as a result of such involvement.[15] However, as well as being a vehicle for improving research validity, there are indications that 'patient influence' can pose a potential threat to the validity of research if it is not drawn on appropriately.[2] For example, PPI in technical decisions may result in worse as opposed to improved project outcomes.[16]

Challenges to the realisation of plans for PPI include debate regarding its purpose, lack of evidence regarding the impact of PPI, complexities in researchers and contributors sharing power, and difficulties in ensuring sufficient resources for PPI.[4 10 15 17–19] Alongside such challenges are uncertainties regarding how best to plan PPI. Guidance drawing on the opinions and experiences of those involved in PPI activity within trials is available[17 20] and a recent review has examined case studies of PPI in the design and conduct of trials.[21] However, the evidence base is limited in terms of the range of trials, researchers and patients that have informed this previous work, and there has been no systematic evaluation of the extent to which trialists' intentions for PPI are put into practice. This is an important gap in view of the above challenges and the increased onus on researchers to build plans for PPI into their grant applications. Such plans run the risk of being uninformed due to the lack of evidence across a range of trial contexts and informant perspectives. In this paper we aim to inform practice for trialists and contributors by describing the extent to which documented PPI plans were implemented within a range of clinical trials and identifying the challenges met and the lessons learnt. Given that funding bodies encourage PPI, we also aim to inform policy with regard to post-trial scrutiny of PPI in terms of processes, facilitators and barriers, and impacts.

## METHODS
### Terminology
We use the term 'PPI contributors' or 'contributors' rather than the more commonly used term 'PPI representatives' to avoid implying that a few individuals can represent the perspectives of diverse patient groups and members of the public, and 'informants' to refer collectively to the researchers (primarily chief investigators (CIs)) and PPI contributors. We use the terms 'documented plans' to refer to the plans for PPI which were written into the funding application or study protocol and 'expectations' to refer to what the trial team expected PPI to achieve, as described by the researchers during the interviews.

### Design
This qualitative study formed part of the 'Evidence base for Patient and public Involvement in Clinical trials' (EPIC) project. EPIC aimed to investigate PPI in a cohort of randomised controlled trials (RCTs) funded by the National Institute for Health Research (NIHR) Health Technology Assessment (HTA) programme between 2006 and 2010. We have described the methods in full elsewhere.[22] In summary, EPIC comprised four phases. Phase 1 examined trialists' plans for PPI as described within their outline and full funding applications. Phase 2 was a questionnaire survey of CIs' and PPI contributors' opinions and activities concerning PPI. Phase 3 involved qualitative interviews with CIs, PPI contributors and trial managers (TMs). Phase 4 examined the role of clinical trials units in identifying and supporting PPI activity in trials.

The current paper draws mostly on data from phases 1 and 3 and, to a lesser extent, phase 2. EPIC had a patient advisory group, consisting of five people with experience of being a patient or a carer, previous PPI contribution in trials and lay review of funding applications and membership of funding panels. The National Research Ethics Service (NRES) advised that EPIC did not require NRES ethics approval; we therefore sought and obtained a favourable ethical opinion from the University of Liverpool Research Ethics Committee (Ref: RETH000489).

### Sampling and recruitment for semistructured interviews
We emailed CIs at the address given on their grant application form. We aimed for a diverse sample of CIs for interview, based on their responses to questions within the CI survey concerning motivations for including PPI and its perceived impact, although we ultimately invited all but three of the CIs who had responded to the survey and expressed an interest in being interviewed. Three CIs were not invited because of delays in responding to the survey. We identified and invited PPI contributors to be interviewed through the CIs, chairs of steering committees and advertisements on PPI websites. Potential informants were sent an email with an information leaflet which included the purpose of the qualitative study.

LD conducted semistructured telephone interviews with informants between April and November 2013, seeking their views and experiences of PPI within their trial. The interviewer had a BSc and MRes in psychology, and previous experience and training of conducting and analysing qualitative interviews. Apart from the recruitment emails, the interviewer had not established a relationship with the participants prior to the start of the study. LD was new to the field of patient involvement in research and sought to maintain an open-minded approach in exploring its implementation in trials. The interviews were audio-recorded, transcribed, anonymised and checked for accuracy. The interviewer used topic guides which were reviewed by our patient advisory

group, and developed in light of ongoing data analysis. The interviews were conversational in nature, enabling informants to freely describe their experiences and raise topics which we had not anticipated. Informants gave their informed consent for the interviews to be audio-recorded and analysed. During the interviews we asked all informants to describe the type of PPI activity that had taken place in the trial. In order to foster rapport between informant and interviewer we intentionally avoided direct questions about why any plans were not implemented. However, we did ask CIs whether they would do anything differently regarding PPI if they were to start the trial again. We asked PPI contributors about any challenges and explored their views on how PPI could be enhanced in future trials. No field notes or repeat interviews were undertaken.

### Data sources

Primary sources of data were: trial documentation (full application forms, reviewer comments, detailed project descriptions and study protocols), from which we extracted data about plans for PPI; CI and PPI contributor interview transcripts, from which we determined whether the documented plans were implemented. Secondary sources of data were: outline application forms, CI survey responses and TM interview transcripts. We used the secondary sources in cases of ambiguity, that is, where it was unclear from the primary sources whether aspects of a particular set of plans had been implemented. We also used the secondary sources to elucidate the illustrative examples that we present in the results below.

### Analysis

To be eligible for the current analysis at least one source of interview data was required from either the CI or PPI contributor, as well as the grant application documents from which we identified and extracted data regarding plans for PPI. To determine the extent to which these documented plans were implemented we focused equally on the qualitative data from the CI and PPI contributor interview transcripts. In cases of ambiguity we consulted the TM interview transcripts, where available. We focused on identifying and analysing patterns within the data, to inform our interpretations,[23] and as appropriate the criterion of catalytic validity whereby qualitative research should not just describe but aim to inform practice.[24] For the purposes of determining the PPI activity undertaken, challenges met and lessons learnt, one author (DB) first familiarised herself with the data by reading the transcripts several times, before drawing on the framework technique[25] to develop and apply open codes to the interview data. She then grouped the codes into broader categories within the framework and compared these with data extracted from the documented plans. Other members of the EPIC team who were familiar with the interview transcripts and documented plans examined the early stages and ongoing refinements of the

descriptive coding framework, as well as the tabulated comparisons of planned and implemented PPI. CG had analysed the CI survey and application forms,[22] and LD and BY had analysed the interview data to explore the perceived impact of PPI, thus providing confidence in the credibility and 'confirmability' of the present findings.[26] Moreover, DB analysed the interview transcripts before looking at the documented plans that had been extracted from the grant application forms, thus helping to reduce the chances that the documented plans would unduly influence her interpretations of informants' interview accounts of PPI. Transcripts were not returned to informants for 'member checking' as interpretation of such feedback is problematic.[27] A description of the coding frame is available on request.

We provide illustrative quotes from a range of interviews and trial documents. Identification codes signify the source of informant quotes based on their group (ie, CI or PPI contributor) followed by their anonymised trial identification number. Where more than one PPI contributor was interviewed for the same trial, we indicate as PPI 1 or PPI 2. Codes for documented plans refer to anonymised trial identification numbers. We replaced identifying text within quotes with anonymised text, and use […] to signify abridged quotes.

In the sections that follow we refer to the three different types of PPI role, identified by our earlier analysis of informants' accounts of the impact of PPI on the trials. The identified PPI roles were: oversight, typically characterised by the formal presence of a PPI contributor on the trial steering committee (TSC), with infrequent involvement; managerial, also usually a formal role but with more regular involvement, for example as co-investigator or member of the trial management group; and responsive roles, which tended to be less formal, often with more than one contributor, or making use of advisory panels and focus groups as and when problems occurred.

## RESULTS
### PPI plans: from intentions to actions

As illustrated in figure 1, 28 trials were eligible for inclusion in the current analysis. We conducted interviews with the CI and a PPI contributor in 9 of the 28 trials, with the CI only in 12 trials and a PPI contributor only in 7 trials. One PPI contributor was involved in two of the trials in this sample, while a further two trials had two PPI contributor interviews. We also conducted interviews with 10 TMs and consulted 1 of these transcripts where there was ambiguity in CI/PPI accounts regarding whether all plans for PPI had been implemented. Interviews lasted 45 min on average. Where multiple sources of interview data were available, for example, from a CI and a PPI contributor, there were no major discrepancies between accounts.

As shown in table 1, all but 3 of the 28 trials had documented plans for PPI in their grant application or

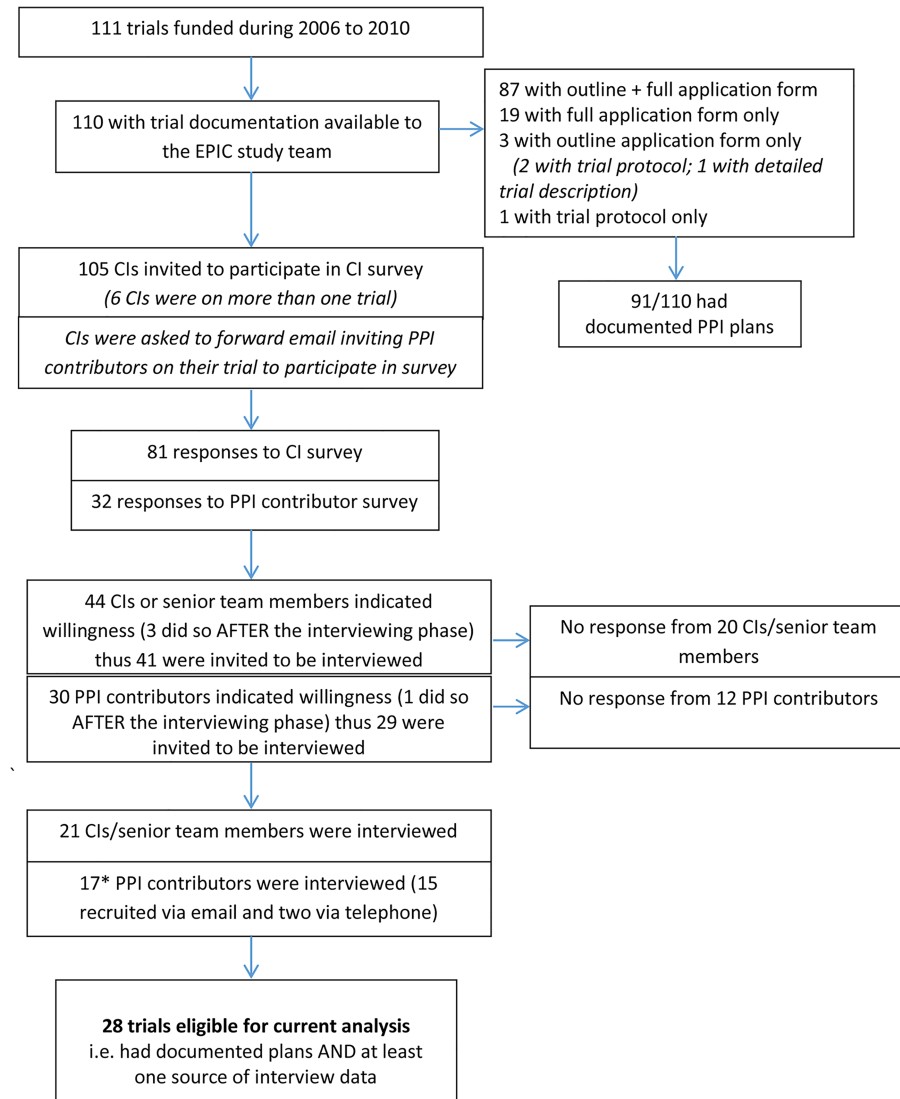

**Figure 1** EPIC trials eligible for analysis comparing PPI plans and implementation. *There were 17 contributor interviews for 17 trials, although 1 PPI contributor was in 2 trials while a further 2 trials had 2 PPI contributor interviews. CI, chief investigator; PPI, patient and public involvement; EPIC, Evidence base for Patient and public Involvement in Clinical trials.

protocol or both. These documents varied greatly regarding the extensiveness of PPI activity planned and precision with which plans were described, from vague references to activities that hinted at PPI, "We will make use of two primary care research networks and an [intervention-specific] research network" (trial 115), to statements that were quite precise, "The [Society] confirmed their willingness to represent their members through steering committee membership […] and to help in the construction of the MREC application and patient information leaflets" (trial 102). On the basis of informants' interview accounts, all trials subsequently incorporated some form of PPI and it was clear from the interviews that documented plans were fully implemented in most (20/25) instances regardless of whether the plans were vague or precise, minimal or extensive. The three trials without documented plans did proceed to include some PPI activity, perhaps prompted, to an extent, by comments from peer reviewers who had remarked on the lack of PPI plans in each case. This is

particularly likely in trial 2. Here, the grant application referred to prefunding PPI and when interviewed the CI spoke of initial 'tokenism' and 'ignorance' about how PPI should work. A further three trials expanded on documented plans, giving a total of six trials which had seen addition or expansion of plans for PPI.

Despite informants indicating that most of the documented plans for PPI had been implemented, some revealed no personal expectations for PPI and spoke of using it as a means of 'ticking the right boxes'. This raises questions about the motivations behind the PPI plans in some grant applications. As noted, we had previously identified three types of PPI roles within our cohort of RCTs: oversight, managerial and responsive,[22] and many trials built into their plans a combination of these roles. On the basis of informants' accounts it appeared that six trials largely confined PPI to an oversight mode of involvement, although some had hinted at other modes in their applications. We begin by examining what happened in these trials.

**Table 1** Summary of planned and implemented PPI activity by type of role

| Trial ID Status (trial ended or ongoing) Mode(s) | Summary of planned activity* | PPI plans fully implemented? | Accounts of 'actual' PPI activity† |
|---|---|---|---|
| (A) Trials which had a chiefly oversight mode (n=6) | | | |
| 115 Ended Oversight | Unclear whether trial had PPI co-applicants although service user contributed to the proposal "We will make use of two primary care research networks and an exercise research network" | U | Had PPI membership on TSC but unclear in terms of 'making use of research networks'. CI had expectations for and prior experience of PPI; no challenges. PPI contributor had prior experience of PPI; challenges (problems getting to meetings because of health) |
| 36 Ongoing Oversight | No PPI co-applicants Patient rep was named as a member of the TSC In response to referee comments, applicants stated they would consider increasing the number of PPI contributors on the TSC from one to two "to provide mutual support" | Y | Has two PPI contributors on TSC but CI talked of 'no direct impact' and 'ticking a political box'. CI had no expectations for but had prior experience of PPI; challenges ("only very minor such as patient rep not having email"). PPI contributor had no prior experience of PPI; challenges (jargon) |
| 65 Ended Oversight | No PPI co-applicants "We will have lay representation on the TSC. We will use the expertise and contacts of our panel to form focus groups to assist in the understanding and dissemination of findings" | U | Had PPI membership on TSC as planned but unclear whether implemented plans regarding the use of the panel/focus groups to understand/disseminate findings. CI felt no direct PPI overall. CI had no expectations for but had prior experience of PPI; challenges (getting the right people engaged; difficult target population; unable to get enough early engagement to inform changes to study design). No PPI contributor interview |
| 2 Ongoing Oversight | No PPI co-applicants No documented plans. Did refer to PPI that had occurred prior to grant application | NA | Has PPI membership on TSC. CI had no expectations for but had prior experience of PPI although spoke of initial "tokenism" and "ignorance" about what to expect of PPI in current trial; challenges ("just the slight feeling that we were taking up her time"). No PPI contributor interview |
| 64 Ended Oversight | No PPI co-applicants "We have identified two people with [condition] who have agreed to be consumer reps and have advised on the development of this proposal" | Y | No CI interview. Had PPI membership on TSC. PPI contributor had no prior experience of PPI; challenges (jargon, unable to attend all the meetings, some team members were felt to lack understanding) |
| 96 Ongoing Oversight | No PPI co-applicants "A patient representative will provide input into the design of patient literature and trial presentations to a general audience as well as providing a patient's perspective at TSC and [Data Monitoring and Ethics Committee] meetings. TSC will meet two to three times a year" | Y | No CI interview. Has PPI membership on TSC. 'Keep in contact' approximately twice a year. PPI contributor had no prior experience of PPI; no challenges |
| (B) Trials which included a managerial mode (n=14)§ | | | |
| 20 Ended Managerial + responsive | Had a PPI co-applicant "The research team will convene a steering group of research and service users. This will meet three times during the study and will provide an opportunity for the research team to consult about research design and methods for data collection, choice of outcomes and methods for data analyses. The TSC will have an important role in interpreting initial findings and developing dissemination strategies. Consultation with young people and parents will be carried out in intervention and comparison clinics using focus groups. The views gathered in these groups will inform the development of research procedures (eg, consent, outcome measures), tools for data collection and the process evaluation. Focus groups will also provide opportunity for young people to contribute to interpretation of study findings. Further consultation with young people will involve piloting all research tools to ensure acceptability and appropriateness" | Y | Had input from four PPI contributors at different times. Membership on TSC. Sought additional input when struggling with particular issues. CI had expectations for and prior experience of PPI; challenges (having a contributor who was a patient of the lead PI—'conflict of roles'; frustration at inability to integrate contributors' ideas regarding questionnaire which was a validated instrument and therefore could not be altered). PPI contributor had no prior experience except as charity member; no challenges |
| 21 Ended Oversight + managerial + responsive | Had a PPI co-applicant "User and consumer groups have discussed the application and suggested changes to protocol which we have accepted. In the trial the groups will be asked to help with development of info leaflets, consent forms, letters, questionnaire design. The groups were very keen that a user was a collaborator on grant application. The team includes [name], a consumer representative who is chair of [Consumer Research Group], works with the [condition] Association and the [Research Network]" | Y | Had PPI co-applicant. Plans expanded (in terms of recruitment, analysis, interpretation of results, dissemination). CI had expectations for and prior experience of PPI; challenges ('poaching' of contributors; stress about funding/paying contributors for their time if in receipt of benefits/pension; disagreement with funders regarding contributor's activities). PPI contributor had prior experience of PPI; challenges (time; being in demand) |

**Table 1** Continued

| Trial ID Status (trial ended or ongoing) Mode(s) | Summary of planned activity* | PPI plans fully implemented? | Accounts of 'actual' PPI activity† |
|---|---|---|---|
| 27 Ongoing Oversight + managerial + responsive | No PPI co-applicants "We will include two [condition] patients to act in an advisory capacity. They will be invited to attend all collaborator meetings and quarterly trial management meetings. We will disseminate project information and findings for patients and patient groups" | Y | Has PPI membership on trial management, steering, and data monitoring groups. CI had expectations for and prior experience of PPI; challenges (finding contributors). Two PPI contributors interviewed had no prior experience of PPI; challenges (some doctors do not want to understand your point of view; jargon; they talk about things you have gone through as a patient in a dispassionate way) |
| 16 Ongoing Oversight + managerial | Had a PPI co-applicant "[Name] is Head of Policy and Research at [name of a national trust]. She has extensive experience of representing the views of the consumer in clinical research and at local and national policy levels. [She] will ensure that the perspective of the consumer remains central during all stages of the trial. Independent user representative(s) will be included on the TSC. The role of user representatives on the Data Monitoring Committee is more difficult because of the complex technical nature of the role of this committee. However, once a Chair of the Data Monitoring Committee has been appointed, we will discuss with the Chair their views about the composition of this committee, and specifically the role of users. User groups at annual [User Group meeting] have commented on the proposal and several groups have agreed to help develop the information and consent process" | Y | Has PPI co-applicant. CI had expectations for and prior experience of PPI; challenges (finding the right people; consumer groups with a specific interest and so may be 'partisan'). PPI contributor had prior experience of PPI; challenges (jargon; infrequent meetings 'not much to build a relationship on') |
| 5 Ongoing Oversight + managerial | Had a PPI co-applicant "We have identified consumer representation from participants in our previous studies, and one, who is a grant applicant, has contributed to the development of the application, trial design and study documentation, particularly the information to be provided about the safety and efficacy of [device]. We have identified a consumer representative to ensure that patients' views are incorporated into the design from the start. She is a grant applicant and has already contributed to the trial design and the participant information sheet. Consumer groups will ensure all relevant issues are covered, that patient information and survey instruments are acceptable and outcome measures relevant" | Y | Has PPI co-applicant. CI had no expectations for but had prior experience of PPI; challenges (finding the right people; finding people without an 'axe to grind'). Two PPI contributors interviewed had no prior experience of PPI; challenges (jargon, not liking flying) |
| 10 Ongoing Oversight + managerial + responsive | Had PPI co-investigator No documented plans | NA | Has co-investigator (from local authority). Consulted with parents regarding timing of intervention. Has a contributor on TSC. When getting low response, approached (education professionals) for advice. CI had expectations for PPI; said had no formal PPI experience 'only informal'; challenges (sometimes difficult to get in touch with co-investigator contributor due to other commitments). PPI contributor had prior experience of PPI; challenges (concern about 'being too pernickety') |
| 4 Ended Managerial | No PPI co-applicants "A project management steering group […] will include all co-applicants, research assistants and user representatives. User representatives will be involved in the development, implementation and interpretation of the study. This involvement will include: advice on recruiting patients, invitation letters, the design of information leaflets, and research instruments, piloting assessments, helping to assess progress, and contributing to the evaluation of the project, the interpretation of findings and the dissemination of results. User representatives will be invited to project steering group meetings and also provide assistance in each centre" | Y | Had 2 PPI members on the trial management group. Involved in most activities as envisaged and while unclear from CI interview about plans for interpretation of the study, responses to the CI survey indicate that analysis had not yet started. CI had expectations for and prior experience of PPI; no challenges. No PPI contributor interview |
| 7 Ongoing Oversight + managerial + responsive | Had a PPI co-applicant "We will include patients and carers as active participants in the research at all stages. [Name] and [name] have taken the role of patient representatives during the preparation of this research proposal. As the relevant service users are highly likely to be frail, we will use innovative methods to allow full involvement. We will not expect attendance at full research team meetings by patients or carers, although our patient representatives may bring their views to the team meetings, following | Y | Has PPI co-applicant and membership on trial management, steering and data monitoring groups. Also consult separate panel of service users for specific issues. CI had expectations for and prior experience of PPI; challenges (identifying/ engaging the right people; some less able to articulate their views; some wanting to do something impossible; difficulty getting other staff to understand or prioritise PPI). No PPI contributor interview |

**Table 1** Continued

| Trial ID Status (trial ended or ongoing) Mode(s) | Summary of planned activity* | PPI plans fully implemented? | Accounts of 'actual' PPI activity† |
|---|---|---|---|
| | meetings with individual or groups of service users in other forums. We identified service users to be involved in this trial through the [names of 2 organisations]. Our named co-applicant will attend Trial Management Group meetings throughout the study in order to contribute the service user perspective at all stages. In addition, [name] is a named co-applicant to the study and will play a role in ensuring that a patient focus is maintained throughout the study. We also plan to seek further views through a wider stakeholder group that will feed into the Trial Management Group through a nominated representative" | | |
| 14 Ongoing Managerial | Had a PPI co-applicant "Co-applicant with an academic interest in representing patients' perspectives in the design and conduct of health care research will advise the research team on the development of processes and materials which take into account patient concerns" | Y | Has PPI co-applicant but CI felt it was a 'tick box' exercise. CI had no expectations for or prior experience of PPI; challenges (meetings attendance; lack of engagement). No PPI contributor interview |
| 41 Ongoing Oversight + managerial + responsive | Had a PPI co-applicant "A representative from [charity] has been involved in preparatory work and will be nominated as a member of the TSC. A minimum of two users will be invited to be part of the project team. A virtual user advisory group will be developed to provide further user support as appropriate. User involvement will contribute to: TSC and project management decisions on all stages of the project; project approval; refinement of self-assessment tools and advice package, exercise intervention; training events for health professionals; interpretation of findings; evaluation of user involvement; dissemination" | Y | Has PPI co-applicant. Trial has 2 PPI contributors although CI feels no strong PPI input overall. Unclear whether CI had expectations for PPI; had no prior experience of PPI; challenges (contributors with an 'axe to grind'; contributors' lack of confidence about contributing at meetings). No PPI contributor interview |
| 55 Ended Oversight + managerial | Had a PPI co-applicant "Patient reps have been very much involved in the preparation of this bid since its inception. The lead service user joined the TSG, will co-ordinate involvement of service users in the consumer panel and report their views to the TSG. Members of the consumer panel have commented on the current proposal and will be asked to comment on specific design and / or management issues during the course of the study. In particular, their views have been, and will continue to be sought during the preparation of patient information leaflets and posters, and in the preparation of study newsletters. They will be asked to help with dissemination of research findings" | Y | Had PPI co-applicant. Planned to involve consumer panel in dissemination of the findings. This did not happen but PPI 'evolved' because the team disseminated through other partners, that is, other patients they were 'working with in the field' by that time. Other plans were adhered to. CI had expectations for and prior experience of PPI; challenges (not realising how much training the panel might need; not being clear about expectations of the main contributor; panel feeling ostracised; difficulty getting trial manager to understand importance and use of the patient panel in the early stages). No PPI contributor interview |
| 15 Ended Oversight + managerial | Had a PPI co-applicant "[Name], a former patient and lay member of the advisory panel, has been fully involved in the application process as a co-applicant and will be a full, active and vocal member. The trial will be guided by a group of respected and experienced critical care personnel and trialists as well as a 'lay' representative" | Y | No CI interview. PPI co-applicant helped to prepare paperwork for funding; also member of TSC. PPI contributor had prior experience of PPI; challenges (jargon) |
| 34‡ Ended Managerial | Had a PPI co-applicant "This proposal has been reviewed by our patient service user group and any opinions and comments incorporated. A patient representative will attend TSC meetings and be directly involved in decision making of trial processes and then relay back information to the [user groups] on a regular basis. Our Service Users group will be involved in all aspects of project design, data collection, analysis and dissemination" | U | No CI interview. Had PPI co-applicant who appears to have been involved as intended, but it is not clear whether plans to involve the user group in data collection, analysis and dissemination were implemented. PPI contributor had prior experience of PPI; challenges (not being involved from the start) |
| 18‡ Ongoing Managerial | Unclear whether had PPI co-applicants Same plans as trial 34 above‡ | U | As above except unclear whether the informant was a co-applicant on this particular trial |

Continued

**Table 1** Continued

| Trial ID<br>Status<br>(trial ended or ongoing)<br>Mode(s) | Summary of planned activity* | PPI plans fully implemented? | Accounts of 'actual' PPI activity† |
|---|---|---|---|
| (C) *Trials which included a responsive role (n=14)§* | | | |
| 20<br>Ended<br>Managerial + responsive | Had a PPI co-applicant<br>"The research team will convene a steering group of research and service users. This will meet three times during the study and will provide an opportunity for the research team to consult about research design and methods for data collection, choice of outcomes and methods for data analyses. The TSC will have an important role in interpreting initial findings and developing dissemination strategies. Consultation with young people and parents will be carried out in intervention and comparison clinics using focus groups. The views gathered in these groups will inform the development of research procedures (eg, consent, outcome measures), tools for data collection and the process evaluation. Focus groups will also provide opportunity for young people to contribute to interpretation of study findings. Further consultation with young people will involve piloting all research tools to ensure acceptability and appropriateness" | Y | Had input from four PPI contributors at different times. Membership on TSC. Sought additional input when struggling with particular issues. CI had expectations for and prior experience of PPI; challenges (having a contributor who was a patient of the lead PI—'conflict of roles'; frustration at inability to integrate contributors' ideas regarding questionnaire which was a validated instrument and therefore could not be altered). PPI contributor had no prior experience except as charity member; no challenges |
| 101<br>Ended<br>Oversight + responsive | No PPI co-applicants<br>"We will convene user group meetings in each locality during the pilot study, we will organise separate focus groups to explore expectations of treatment. We have a commitment from panels of users/experts including representatives from relevant charities to meet annually during the study to advise on its conduct. We have lay representation on the TSC" | Y | Had PPI membership on TSC and consulted with wider groups as planned. CI felt PPI was under utilised and said "people above me in the scheme of things may see it as a tick box exercise." CI had no expectations for PPI; unclear regarding prior experience of PPI; challenges (finding suitable people, 'pinning people down', some may find it daunting whereas 'professional PPI reps' do not). PPI contributor had prior experience of PPI; no challenges |
| 21<br>Ended<br>Oversight + managerial + responsive | Had a PPI co-applicant<br>"User and consumer groups have discussed the application and suggested changes to protocol which we have accepted. In the trial the groups will be asked to help with development of info leaflets, consent forms, letters, questionnaire design. The groups were very keen that a user was a collaborator on grant application. The team includes [name], a consumer representative who is chair of [Consumer Research Group], works with the [condition] Association and the [Research Network]" | Y | Plans expanded (in terms of recruitment, analysis, interpretation of results, dissemination). CI had expectations for and prior experience of PPI; challenges ('poaching' of contributors; stress about funding/paying contributors for their time if in receipt of benefits/pension; disagreement with funders regarding contributor's activities). PPI contributor had prior experience of PPI; challenges (time; being in demand) |
| 27<br>Ongoing<br>Oversight + managerial + responsive | No PPI co-applicants<br>"We will include two [condition] patients to act in an advisory capacity. They will be invited to attend all collaborator meetings and quarterly trial management meetings. We will disseminate project information and findings for patients and patient groups" | Y | Has PPI membership on trial management, steering, and data monitoring groups. CI had expectations for and prior experience of PPI; challenges (finding contributors). Two PPI contributors interviewed had no prior experience of PPI; challenges (some doctors do not want to understand your point of view; jargon; they talk about things you have gone through as a patient in a dispassionate way) |
| 10<br>Ongoing<br>Oversight + managerial + responsive | Had PPI 'co-investigator'<br>No documented plans | NA | Consulted with parents regarding timing of intervention. Has a contributor on TSC. When getting low response, approached (education professionals) for advice. CI had expectations for PPI; said had no formal PPI experience 'only informal'; challenges (sometimes difficult to get in touch with co-investigator contributor due to other commitments). PPI contributor's challenges: concern about 'being too pernickety' |
| 9<br>Ended<br>Oversight + responsive | Unclear whether there were PPI co-applicants<br>"The TSC will include a patient representative, [name], who has acted in this capacity in several other large-scale trials and is aware of issues that might be raised from the lay perspective. The patient information leaflet and consent form have been reviewed by potential service users, and their comments taken into account in finalising these documents prior to submission for ethics approval" | Y | Unclear whether CI had expectations for or prior experience of PPI; no challenges. No PPI contributor interview |

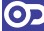

**Table 1**  Continued

| Trial ID Status (trial ended or ongoing) Mode(s) | Summary of planned activity* | PPI plans fully implemented? | Accounts of 'actual' PPI activity† |
|---|---|---|---|
| 102 Ended Oversight + responsive | No PPI co-applicants "At the outline proposal stage, this trial was submitted to the [name of funding body] who sought the opinion of the [condition] Society. The [condition] Society unequivocally confirmed their support of the proposed trial. The [condition] Society have also confirmed their willingness to represent their members through steering committee membership of the [name of trial] and to help the trialists in the construction of the MREC application and patient information leaflets" | Y | Seems to have expanded plans (in terms of dissemination, ie, press releases and findings for participants). CI had expectations for and prior experience of PPI; no challenges. No PPI contributor interview |
| 6 Ongoing Oversight + responsive | No PPI co-applicants "The TSC will include an already identified patient. He will provide an informed patient perspective. He is willing to assist us in the trial, and will be listed as a member of the TSC. We will also work with [charity] to involve service users. This will be done through our links with the [unit], which is co-directed by one of our applicants, [name]. We will begin this process during the protocol set-up period" | Y | CI had expectations for but unclear whether had prior experience of PPI; no challenges. No PPI contributor interview |
| 7 Ongoing Oversight + managerial + responsive | Had a PPI co-applicant "We will include patients and carers as active participants in the research at all stages. [Name] and [name] have taken the role of patient representatives during the preparation of this research proposal. As the relevant service users are highly likely to be frail, we will use innovative methods to allow full involvement. We will not expect attendance at full research team meetings by patients or carers, although our patient representatives may bring their views to the team meetings, following meetings with individual or groups of service users in other forums. We identified service users to be involved in this trial through the [names of 2 organisations]. Our named co-applicant will attend Trial Management Group meetings throughout the study in order to contribute the service user perspective at all stages. In addition, [name] is a named co-applicant to the study and will play a role in ensuring that a patient focus is maintained throughout the study. We also plan to seek further views through a wider stakeholder group that will feed into the Trial Management Group through a nominated representative" | Y | Consulted separate panel of service users for specific issues. CI had expectations for and prior experience of PPI; challenges (identifying/engaging the right people; some less able to articulate their views; some wanting to do something impossible; difficulty getting other staff to understand or prioritise PPI). No PPI contributor interview |
| 41 Ongoing Oversight + managerial + responsive | Had a PPI co-applicant "A representative from [charity] has been involved in preparatory work and will be nominated as a member of the TSC. A minimum of two users will be invited to be part of the project team. A virtual user advisory group will be developed to provide further user support as appropriate. User involvement will contribute to: TSC and project management decisions on all stages of the project; project approval; refinement of self-assessment tools and advice package, exercise intervention; training events for health professionals; interpretation of findings; evaluation of user involvement; dissemination" | Y | Has PPI co-applicant. Trial has two PPI contributors although CI feels no strong PPI input overall. Unclear whether CI had expectations for PPI; had no prior experience of PPI; challenges (contributors with an 'axe to grind'; contributors' lack of confidence about contributing at meetings). No PPI contributor interview |
| 79 Ended Oversight + responsive | No PPI co-applicants No documented plans | NA | Although no documented plans, the CI wanted PPI to sit on TSC and comment on patient info leaflets. The CI felt that PPI started early. There were two types of involvement: two contributors on the TSC; and then obtained views on information sheets from relevant groups. CI had no previous experience of PPI; no challenges. No PPI contributor interview |

Continued

**Table 1** Continued

| Trial ID Status (trial ended or ongoing) Mode(s) | Summary of planned activity* | PPI plans fully implemented? | Accounts of 'actual' PPI activity† |
|---|---|---|---|
| 76 Ongoing Oversight + responsive | No PPI co-applicants "The [organisation] has recently established a Research Advisory Group. This Group, which includes key stakeholders with an interest in the research carried out by [organisation] (patients, charities representing patients' interests, general practitioners, NHS commissioners, research funding organisations and a regional [medical] network), has been set up to ensure that the clinical research carried out in [organisation] is ethical, important, relevant, appropriately designed to meet the needs of patients and the NHS. We anticipate the Group would have the opportunity to influence important details of the project before recruitment starts. A patient representative (we propose a member of the [advisory group]) will be invited to join the TSC" | U | Has PPI membership on TSC as planned; unclear whether plans to seek advice of new advisory group prior to recruitment were implemented (although did approach a group of patients from a previous trial about format/comprehensibility of questionnaire). CI talked of a "tick box exercise" but also ensuring participants' perspective; "overseeing the trial—a 'safeguard' rather than improving research." CI had expectations for but no prior experience of PPI; challenges (communication and understanding). No PPI contributor interview |
| 106 Ended Oversight+responsive | No PPI co-applicants "We have consulted widely, including with patients to seek their views on trial design and relevant outcome measures. We have involved service users in the design of the trial. We used the patient information pack and part of the questionnaire that has been developed and validated in collaborative research with the [institute] as a basis for in-depth interviews to identify patient perspectives on trial design and outcomes. We have identified one service user, [name], who will advise the trial management committee on patient perspectives" | Y | No CI interview. PPI contributor had prior experience of PPI but felt she had made no difference to the trial; no challenges |
| 91 Ongoing Oversight+responsive | No PPI co-applicants "We have involved [name] who is a non-executive patient representative member of [hospital trust] and who has co-ordinated consumers' input into the scientific quality, feasibility and practicality of the proposal. She will continue to participate in the protocol design of the study and be a member of the TSC" | Y | No CI interview. Plans expanded (in terms of the PPI contributor obtaining feedback from "women's groups"). PPI contributor had prior experience of PPI; challenges (just being confident enough to make your point) |

☐ Based on informants' accounts, it was unclear whether the trial fully implemented or was implementing all plans.

☐ Based on informants' accounts, the trial did what planned in that PPI had been or was being fully implemented.

☐ NA no documented plans.

*As described in the funding application and/or study protocol; includes justification of costs where data were available.

†As reported during informant interviews—any reference to tokenism; whether CI had prior experience of or personal expectations for PPI; whether CI mentioned challenges; whether PPI contributor mentioned challenges.

‡PPI contributor was discussing two trials (ID 18 and 34) during the interview.

§Many trials utilised more than one form of PPI.

CI, chief investigator; PPI, patient and public involvement; TSC, trial steering committee; U=unclear; Y= yes.

## Oversight mode trials (n=6)

Oversight mode trials were those which confined PPI input to membership of TSCs. On the basis of informant interview accounts, there were six trials that constrained PPI to this mode of involvement, although three of these had hinted at other modes in their applications. A further application had been too vague to discern the mode of planned PPI, and another had no documented plans for PPI (table 1).

On the basis of informants' accounts, all trials which had documented plans for PPI membership on their TSC had implemented this aspect of the plans. Researcher interviews were available for four of these six oversight trials and of the four, only one researcher divulged any personal expectations for PPI in the trial. Moreover, informants' accounts raise concerns about the motivations for including PPI in their applications and the danger of assuming that contributors know what is expected of them. For example, trial 36 had named a 'patient representative' as a member of the TSC at the application stage then subsequently, in direct response to peer reviewer comments, the team had indicated that they would consider increasing the number of 'patient representatives' on the TSC from one to two, in order to provide 'mutual support'. The team proceeded to include two PPI contributors on the TSC, thereby achieving their documented plans. Despite having prior experience of PPI, however, the researcher divulged no personal expectations for PPI within this particular trial and referred to PPI as a 'tick box' exercise:

> It was a requirement of…that we had representation on our steering committee and therefore I went through that […] We can say [the PPI contributors] are there and therefore it's, if you like, ticking a political box. (CI 36)

The documentation for trial 2 included no plans for PPI during the trial but did state that there had been 'several stages of user involvement' prior to the grant application, "to confirm that the research question is pertinent to both the needs of the NHS and the NIHR programme of research development." Two grant reviewers commented on the lack of 'service user representation' on the team and suggested membership 'on the research team or steering group'. The TSC did include PPI membership but during the interview the researcher spoke of his initial 'tokenism' and 'ignorance' about how PPI 'should and could work'. When asked about the expectations of their role, the PPI contributors in two other oversight trials (115 and 96) implied similar uncertainties when they spoke of not knowing what was expected of them and of feeling 'bewildered' in meetings:

> I can't understand why they use me… they seem to find me useful but I just sit there bewildered. I'm there as a sort of grey background while the others do all the sparky stuff. (PPI 115)

In the next section we describe planned and implemented PPI in 14 trials which incorporated a managerial role of PPI. Unlike the six trials with a mainly oversight mode, many of the managerial mode trials had utilised more than one form of PPI.

## Beyond oversight, into managerial mode (n=14)

Most of these 14 trials had indicated some type of managerial involvement in the documented plans, usually to include PPI contributors as co-investigators (table 1). Two trials (4 and 27) did not have PPI contributors as co-investigators but planned to include PPI contributors on the trial management group, and interviews with informants indicated that this had been implemented. It was unclear in one ongoing trial whether there was a PPI co-investigator, but documented plans stated that a named PPI collaborator would be "directly involved in decision making of trial processes and then relay back information to user groups"; according to the PPI contributor interview these plans were being implemented (trial 18). Trial 10 had no documented plans for PPI but the interview with the CI indicated that there was a PPI co-investigator (trial 10).

Informants' accounts indicated that all trials which had planned a managerial mode of PPI did implement it (table 1). This included trial 21, which had a PPI co-applicant and documented plans to involve user groups in developing information leaflets, consent forms, letters and in questionnaire design. There was a budget for PPI travel and expenses which is perhaps indicative of careful planning. The documented plans stated that "user and consumer groups were very keen that a user was a collaborator on the grant application." The applicants also planned and included oversight PPI (TSC membership) and expanded beyond their plans to include contributors in recruitment, in the analysis and interpretation of results, and in dissemination. Although we could not pinpoint from the informant interviews exactly what prompted these additional PPI activities, the PPI contributor who we interviewed described his extensive previous experience in similar roles and noted that his role in this particular trial had 'evolved'. He also explained that "I'm there because I want to change things" (PPI 21) and this proactive approach may have contributed to the expansion of PPI in this particular trial. Correspondingly, the CI spoke of wanting the PPI contributors to "feel welcomed and valued as part of the group," and had personal expectations for PPI that included PPI contributors helping with 'running the study', 'disseminating the results' and that 'they would stay involved' and 'feel able to speak out and have their own opinion':

> We wanted them to offer to do things that they felt they could do and feel happy to say if they didn't feel they could do certain things that might come their way. (CI 21)

There were several examples akin to this among trials incorporating a managerial mode of PPI, in which CIs

reported having personal expectations for PPI or in which PPI contributors appeared to be an integral member of the research team. However, one of the two exceptions was trial 14, in which documented plans had been to involve a PPI co-applicant "with an academic interest in representing patients' perspectives in the design and conduct of health care research," adding that this individual would advise on "the development of processes and materials which take into account patient concerns." Responses to the CI survey described the PPI contributor as 'a serial patient representative'. When interviewed, the CI divulged no personal expectations regarding PPI contribution, describing it as a 'tick box exercise':

> The funders were insistent on having patient representation and wanted to know what that representation was on your grant submission. (CI 14)

In summary, most trials which planned a managerial mode of PPI implemented it. However, as trial 14 shows, simply having a PPI co-investigator is not necessarily a guarantee of meaningful contribution if researchers have no expectations for PPI or if contributors are unable to provide the input that a particular trial requires, for example because they are selected out of convenience rather than to match trial needs. In the next section we focus on the less formal, responsive, form of PPI in which researchers 'reach out' for specific PPI input as and when needed.

### 'Reaching out'—responsive roles (n=14)

Fourteen trials embraced some form of responsive involvement, although trial documents for two (10 and 79) had not indicated any plans for PPI (table 1). The remaining 12 had stated in their documented plans that they would, or already did, engage with PPI groups or panels rather than just with the one or two individuals that was typical of oversight and managerial PPI. Data from application forms, project descriptions and informant interviews showed that this responsive activity sometimes entailed seeking advice from PPI groups prior to the application for funding. Informants noted that many trialists continued to seek advice from such groups during the trial regarding specific issues. Other trials began a responsive approach once the trial had begun, often as and when particular problems arose. Most trials implemented all aspects of their documented plans but in one case (trial 76) it was unclear from the CI interview whether specific plans to seek advice of a new advisory group before recruitment were implemented.

Trial 20 used responsive alongside managerial PPI, including having a PPI co-applicant. The trial had ended at the time of the interviews, and the researcher stressed that the responsive PPI had been 'crucial' when faced with specific problems. The CI explained that one PPI contributor would attend research team meetings:

> but I then reached out to other people in addition when we needed more help […] I think what was crucial was being able to get input, not in terms of regular intervals but […] when you've got a problem. (CI 20)

Further illustrating the flexibility that responsive PPI allows, in her interview one of the PPI contributors on the same trial (who on this particular trial had a managerial role), advised researchers to 'have some understanding' of the needs of PPI contributors. She then went on to refer to another contributor on the same trial who did not attend project meetings but who operated in a more responsive mode outside of meetings. It appeared this arrangement had evolved to accommodate the needs of the latter contributor, who, it seemed, found meetings difficult.

> She didn't really know what to do, so I think it was much more a one-to-one conversation which is what she was happy with rather than sitting in a committee. (PPI 20)

Documented plans for trial 7 involved a combination of oversight, managerial and responsive modes. This trial was collecting outcome data at the time of the researcher interview, and PPI plans were being implemented including consultation with a panel of service users who advised on issues such as how to increase participant response rates to the outcome questionnaire, and on the promotional material that accompanied it. When interviewed, the researcher spoke of her personal expectations that PPI would help to maximise recruitment, ensure the right outcomes were measured, and help in interpreting the findings. There was no PPI contributor interview but the researcher also spoke of having to tailor 'different ways of involving people' in PPI depending on the 'population of interest':

> It might be children, people from disadvantaged groups or older people […] so you probably have to find other tailored ways of including people to make it effective. So it's not a one size fits all. (CI 7)

The majority of those researchers interviewed who described such 'as and when' contributions (10/12) spoke of expectations for PPI, and tended to view responsive modes as constructive. Only in one case (trial 101) did the researcher allude to the PPI within their trial as a 'tick box' exercise.

Three trials undertook additional responsive PPI activity that had not been specified in their documented plans. Trials 21 and 102 expanded on their plans by involving PPI contributors in a broader range of activities than initially indicated, namely advising on recruitment and interpretation and dissemination of study findings. As with trial 21 (described in the Managerial Mode section above), we could not determine from the CI interview why plans for Trial 102 had been expanded, and there was no PPI contributor interview for trial 102 to help illuminate this issue. The PPI contributor for the

**Table 2** Summary of challenges met by CIs and contributors to PPI in clinical trials

| CI interviews (n=21) | PPI contributor interviews (n=17)* |
|---|---|
| Challenges common to researchers and PPI contributors: | |
| Failure to engage contributors fully or early | Not being involved from the start; Infrequent meetings |
| Contributors overawed/lacking confidence | Feeling unqualified or overwhelmed |
| Failing to clarify to contributors what was expected of them | Role expectations (being unsure what was expected of you) |
| Worry about taking up contributor's time | Time constraints |
| Contributors being 'poached' | Being in demand by other research teams |
| Meeting attendance by PPI contributors | Getting to meetings |
| Challenges unique to researchers or PPI contributors: | |
| Finding the right people | Jargon |
| Own patient as a PPI contributor (can lead to conflict between clinical and research roles) | Interactions within team and being listened to |
| Communication difficulties due to age | Concern about appearing confrontational |
| Change of PPI personnel | Concern about appearing too 'pernickety' |
| Getting other team members to understand/prioritise PPI | Remembering 'what side you are on' |
| Underestimating training needs of contributors | |
| Worry that contributors may lose payment if receiving state pension/benefits | |
| Disagreement with funders about implementing contributors' suggestions | |

*One PPI contributor was involved in and talked about two trials which were in this sample, and there were two trials for which we had two PPI contributor interviews each.
CI, chief investigator; PPI, patient and public involvement.

third trial (trial 91) mentioned that she sought the views of 'women's groups'. This was additional to the documented plans for her to be involved in 'protocol design of the study'. As with Trial 21, this PPI contributor had previous PPI experience and appeared to be a particularly active member of the research team, and with considerable knowledge of the relevant health condition.

In summary, most applicants implemented their documented plans for PPI regardless of the mode of planned involvement. In five cases we were unable to discern whether or not PPI plans were fully implemented, although some PPI was achieved in these trials. Regardless of whether PPI was implemented as planned or evolved, most trial teams faced challenges and learnt lessons about implementing PPI as they went along. We now turn to their accounts of this learning and then use these to derive practical advice for planning and implementing PPI.

### Researchers on the challenges of PPI and lessons learnt

Most CIs spoke of the challenges they encountered in implementing PPI (table 2) and things they would do differently as a result. The involvement of trial investigators' own patients as contributors was perceived to lead to a 'conflict' (CI 20) between an investigator's research and clinical roles. This brought a risk that research would "cross over into clinical care" (CI 6), and that such contributors would be 'out of their depth' (CI 20) and find it difficult to "say something which might imply a criticism of their clinician" (CI 20). CIs talked about the problems of failing to engage PPI contributors fully

or early enough to inform changes in study design, and 'under-utilising' (CI 101) PPI contributors by not involving them in the planning stages, thereby making PPI less thorough or, as one informant noted, less 'robust' (CI 101). They reflected on the potential detrimental consequences of such failings on the relationship between researcher and PPI contributors, for example being less likely to "form a bond and get loyalty" (CI 14). Finding and engaging the right people with an interest in and understanding of the research, and with the necessary confidence, commitment and impartiality was another major stumbling block:

> You hear that some consumers get involved […] because they have a particular point of view or axe to grind […] in those circumstances it could be very detrimental to a trial, to be driven by somebody who has had a bad experience […] and those are the ones you don't want on your team. (CI 5)

> You've got trialists in the [meeting] who are trained to run clinical trials. And then you've got one lay representative who may be slightly intimidated by everyone else, who'll not be able to truly give their views, may be slightly overawed. (CI 14)

Researchers also pointed to the practical difficulties that contributors experienced in attending meetings due to geographical distance or time constraints (table 2). They emphasised how teleconferences could be less conducive to forming a relationship with PPI contributors

than face-to-face meetings. They also reported problems relating to communication and mutual comprehension between themselves and PPI contributors. Some described PPI contributors as struggling to understand the nature of research, or the distinction between research and clinical practice, and one CI referred to his own 'naivety' (CI 55) in underestimating how much training PPI contributors might need. CIs described difficulties getting other staff such as TMs to understand or prioritise PPI. This included one CI who noted that some investigators are unable to 'cope' with having a "working relationship with service users" and "can't let go of the fact that [they] are people they study":

> It's a mindset […] an attitude where you have an equal partnership. You're working together not studying these people. You're asking for their expertise and I've found that some people who've worked with me, that comes easily and some people absolutely never get it. (CI 20)

CIs remarked that they were unclear about what to expect in relation to PPI and worried about taking up the contributor's time. External forces also played a part in some cases: for example, one CI described PPI contributors being 'poached' by other studies, a 'fight' with the university regarding paying a PPI contributor for his time, and disagreement with funders when a contributor wanted to add to the patient information sheet that he was a PPI contributor on the project (CI 21).

CIs spoke of how they had learnt as the trial went along, revealing that their 'practice had evolved' (CI 14) and their skills had "changed beyond recognition […] now we're much better equipped […] but at the time when [trial] started we had very little idea at all about what PPI involved or how it would help or how it would work" (CI 2).

In light of these challenges, CIs spoke of how in future they would involve more than one PPI contributor, in particular by using focus groups or panels of contributors rather than individual contributors, enlist the help of relevant charities, and conduct surveys or use social media when there was a 'burning question' (CI 55). Use of responsive PPI rather than individual contributors was described as 'gold standard' PPI (CI 14), as this avoided "the danger of having a single opinion" (CI 76), provided structure for all parties, and helped to enhance the confidence of individual contributors.

> I would certainly have more involvement and some kind of framework around it […] a small user group and set boundaries […] try to agree how often we should meet and what peoples' roles and responsibilities are […] and provided more structure […] to make them feel that their views are important, and their involvement is very important, I think that would go a long way to easing the process. (CI 41)

Many CIs indicated that they would extend PPI in future by asking contributors to lead in the dissemination of findings to relevant groups, help in the development of research questions, study design, and involve PPI contributors as co-investigators. CIs placed particular emphasis on how 'crucial' it was to have 'early input' (CI 14):

> The most useful things are […] the design stage […] RCTs you've got to plan ahead [...] after the development phase you shouldn't really be changing anything […] it is during that development phase when decisions are being made. (CI 115)

> Early engagement and appreciation that their input into the question is really important […] with retrospect and for the future studies […] more involvement at the front end, less in the middle and more at the end. (CI 2)

Finally, CIs reflected on the importance of 'thinking through' plans and being clear about whether, what and why PPI is needed for individual trials:

> Be clear about the link between particular methods [of PPI] and particular benefits and challenges […] it's not all the same, there are so many ways of doing it but you have to have good reasons for choosing how to do it. (CI 20)

> "I don't think it should be automatic that there must be PPI involvement in every study, and different types of involvement are necessary for different parts of study. Having a core group is not necessarily the right thing because at different points there are different types of people and types of involvement that would be useful. (CI 10)

### Contributors on the challenges of PPI and suggestions for improvement

Most PPI contributors mentioned challenges or difficulties linked to their involvement in the trial which may inform future research teams in planning and implementing PPI. Some of the contributors' challenges paralleled CIs' accounts while others were unique to the contributors (table 2). While researchers referred to problems they had experienced in their communication with contributors, a prominent issue exclusively mentioned by contributors related to the problems they experienced with 'jargon' and the technical language that was used in trials such as statistical or medical terminology and acronyms. Several contributors suggested remedies such as supplying a list of acronyms or a booklet of research terms, or simply that "if they're going to use jargon, explain it" (PPI 64). A further idea was that the person chairing meetings could try to ensure that discussion about statistical issues or other areas of technical expertise were translated and summarised adequately. Contributors talked about difficulties in interacting with researchers, including not always feeling listened to by everyone. One contributor who had been invited by her consultant and had previous experience of PPI implied that 'some doctors' were unwilling to understand the perspectives of patients (PPI2 27). Another felt that female researchers were more understanding than males regarding problems with travelling or feelings of insecurity, while a further contributor alluded to how in meetings

the team sometimes talked about patient experiences in a 'dispassionate' way, and although this was not a problem for the individual contributor she felt it might be for others (PPI1 27).

Some of the challenges that contributors described echoed those that the CIs has raised. These included lack of clarity about roles, and the difficulties contributors experienced in attending meetings, for instance because of a health condition. Such practical difficulties could give rise to additional complexities. For one contributor, infrequent meetings meant "not much to build a relationship on" and while academics worked closely together, she had to "work quite hard to keep up" (PPI 16). Contributors also talked about wanting to be more involved in between annual meetings, in 'shaping the bid' (PPI 20) so that it was less focused on the primary clinical outcome, in seeing the intervention itself, and to have initial briefing meetings at the outset of their involvement. Finally, one contributor described it as a 'downfall' that he was not receiving feedback or 'thank yous' and commented on how important it was to make PPI contributors 'feel valued' (PPI 34).

## DISCUSSION
### Main findings
#### The path to PPI: plans, actions and complications

This is the first study to examine whether plans for PPI, as documented in RCT grant applications, are being implemented. On the basis of the accounts of researchers and PPI contributors we found that most trialists are indeed putting their plans to action, although in some cases the plans were minimal and relatively easy to execute. There were a few trials for which we were unable to confirm whether plans were implemented in full, but all did incorporate some PPI. Many trials implemented multiple modes of PPI, which is surprising and encouraging given that PPI was less prominent when the proposals for the trials in this cohort were being developed. CIs encountered complications from which they learnt valuable lessons. Uncertainty about what to expect of PPI and emergent challenges with their trials meant that involvement had to evolve. Difficulties finding and retaining suitable contributors and engaging in PPI 'too little too late' led trialists to say they would do things differently in future. Many reflected on how they would aim for earlier engagement next time and seek involvement from a more diverse source such as patient panels or focus groups. PPI contributors themselves mentioned that becoming involved after the trial had begun, or infrequently, resulted in missed opportunities for them to contribute. Some referred to uncertainty about their role and many struggled with jargon, an enduring problem despite the availability of apparently straightforward solutions.

#### Pressured into PPI?

Regardless of statements about PPI in their funding application some trialists had no expectations of what PPI might achieve, and their only motivation for including PPI was a belief that it was necessary or would help to secure funding for their trial. Such strategic minimalism may be an inevitable side-effect of policies to promote or require PPI in trials. It may also reflect researchers' professed inexperience of PPI. A small number of trials did not have documented plans for PPI but all did nevertheless include some PPI, possibly influenced by reviewer and panel comments. However, one of these trials had been through several stages of PPI prior to the grant application and was requested to implement further PPI over the course of the trial. This highlights the predicament of researchers whose trial may have benefited from considerable PPI prior to funding (eg, in feasibility and pilot work) and forecast that they would need relatively little PPI during the trial itself, only to find that funders insist on PPI at all stages. Many informants believed formative PPI prior to funding was one of the most useful, credible aspects of PPI. Particularly in cases where there has been extensive PPI prior to the main trial, it is important for all members of the research community to consider whether plans for ongoing PPI match the needs of a particular trial and at what stage(s) further PPI would be appropriate.

### Previous research

We found no previous reports on the extent to which documented plans for PPI within trials were subsequently implemented. There have been several accounts of challenges involved in implementing PPI which, while not in a trials context, endorse our findings. For instance, recent reports have referred to tokenism,[28 29] or highlighted the potential challenges in identifying suitable individuals who are impartial and able to understand research methodologies, retain an interest, and commit long-term;[15 17–19 30] of researchers having little experience of PPI and being uncertain about what to expect;[15 18 31] and of jargon-related problems.[19 32 33] INVOLVE suggest that PPI contributors would benefit from a 'glossary of technical terms',[17] again something reflected in the suggestions from contributors within our study. Staley[4] refers to the challenge of ensuring that involvement is meaningful and not simply tokenistic. Findings from the EPIC project regarding PPI training needs suggest that while informants were broadly receptive to PPI training for researchers, there was considerable reluctance regarding the training of PPI contributors, with a preference for 'informal inductions'. The health services researchers in a previous qualitative interview study varied in how they interpreted PPI policy and in their PPI 'working practices' and referred to how PPI brought a 'fear of the unknown'.[31] This study also points to a 'know-do' gap, whereby researchers' talk of the importance and value of PPI in the 'ideal' world stood in contrast to their experiences of 'the reality' of implementing PPI in practice.[29] The timing of involvement has been recently highlighted[3 20] and is clearly an ongoing challenge which is exacerbated by financial and time constraints[8 32] particularly during the grant-writing stage.

## Study limitations

We used a historical cohort of trials that had been funded 4–8 years ago. Even in that short time the emphasis on PPI has grown and our findings may not reflect the planning and implementation of PPI in trials funded more recently. Some of the trials in our sample were also initiated and completed some time before the interviews. However, this limitation is offset somewhat by the inclusion of ongoing trials in which PPI activity was recent and therefore easier to recollect. There were five trials for which it was not possible to determine whether all documented PPI plans had been fully implemented or not. In some cases informants clearly struggled to recall events for trials which had ended several years previously or where researchers were involved in a number of trials simultaneously. We explored with informants how PPI contributors were involved in the trials but did not directly quiz CIs about why certain plans within their application were not implemented. This was intentional as we did not want to pose questions which may have seemed accusatory and have a detrimental impact on the rapport between informant and interviewer or risk informants becoming defensive. While some trialists seem to have expanded on their plans for PPI once the trial was underway there may, conversely, have been instances in which plans were not fully documented within the grant application.

## Implications and tips for the trials community

We have used the insights of informants to generate practical tips which may help future trialists and PPI contributors (box 1). We envisage that these be considered alongside previously published guidance for PPI in trials[17 20] and consensus principles for PPI in health research.[34 35] The tips generated from evidence in our study cover the importance of early planning, of timely and flexible PPI, and of communication and clarification of roles. They also stress the need to consider the difficulties posed by the use of 'jargon', and problems contributors experience in understanding certain aspects of the research process. The difficulties contributors experience with specialist or technical terminology have been widely reported.[19 32 33] Our data suggest that this problem has existed for some considerable time, and we outline the practical solutions suggested by PPI contributors. The tips in box 1 could be used to inform PPI training and could be helpful in other types of health research. Given that the usefulness of the points in box 1 depends on researchers' willingness to genuinely engage with PPI, the tips we present might also assist funding bodies and grant reviewers in determining whether submitted plans are fit for purpose. A study of the UK health and social care research community has recently informed the development of a Public Involvement Impact Assessment Framework (PiiAF), which emphasises the value of well thought-through planning before implementing PPI as well as the subsequent evaluation of its impact,[36] and

INVOLVE[17] have emphasised the importance of clear guidance about roles. However, researchers also need some scope for flexibility and contingency in planning PPI: our finding that some trialists expanded their sometimes already detailed plans supports the need for flexible and iterative approaches to PPI in order to accommodate the unexpected and respond to opportunities and difficulties as they arise.

## Ticking several boxes could equate to expensive token gestures: implications for funders

Our findings endorse recent revisions to the NIHR's standard application form, which now require applicants to clearly define their proposed PPI activity. Asking researchers to specify and explain the type of involvement they envisage and what they expect it to achieve is a step in the right direction and should help to minimise 'tick box' tactics and token gestures. However, the risk of strategic minimalism remains if plans are not afforded careful, context-specific consideration by funders and reviewers. Equally, there is a risk of inadvertent PPI profligacy, that is, the encouragement of elaborate plans for PPI that are disproportionate to the needs of a trial. Ticking several boxes rather than just one box could equally be a token gesture, as well as an expensive one. Therefore, researchers might be encouraged to think just as much about *why, how* and *when* PPI will be useful, as about *what* and *how much* PPI.

Researchers are also now asked to describe, in their grant applications, any PPI activity that they have undertaken prior to submitting the application. Funding is available to support preapplication PPI, for example the UK-based NIHR Research Development Service offers very small grants, which others have found to be helpful.[37 38] However, these grants are not easily or quickly accessible, particularly for those working to the typically tight deadlines of funding calls. Paradoxically, this renders preapplication PPI the most difficult to implement, even though our findings indicate that it is often most useful at this stage. Innovative organisations that involve patients at a meta-trial level in research priority setting http://www.lindalliance.org/Patient_Clinician_Partnerships.asp and in schemes such as COMET (Core Outcome Measures in Effectiveness Trials)[39 40] which promotes the involvement of patients in developing 'core outcome sets', are providing knowledge and resources that individual trials can use. However, at the level of individual trials infrastructural support for early PPI is also needed. While there have been innovations in this area, for example the US-based Patient-Centred Outcomes Research Institute has recently announced a number of 'Pipeline to Proposals' Engagement Awards,[6] such moves are relatively novel, and similar steps by other organisations would be beneficial. As well indicating the need for structures and resources to support PPI, our findings point to the importance of PPI that is fit for purpose, realistic and proportionate. We found that trialists who fully implemented a primarily oversight mode of PPI perceived little value in this involvement

**Box 1** Tips for planning and implementing patient and public involvement (PPI) in clinical trials

*Early PPI*
"You've got to plan ahead"
▶ Begin planning PPI and consulting with contributors when starting to plan the trial.
▶ Consider including PPI contributors in managerial roles for example, as co-investigators.
Researchers and PPI contributors emphasised how early and regular involvement allowed contributors to input more effectively. PPI prior to the trial (eg, in contributions to grant writing, trial design, feasibility studies) was a key aspect of PPI, and in some cases the most important one.

*Flexible PPI*
"One size does not fit all" "Reaching out was crucial"
▶ Consider whether oversight PPI (eg, on a trial steering committee) is sufficient to meet trial needs.
▶ Involve more than one or two PPI contributors, more than once or twice a year.
▶ 'Reach out' and make use of multiple modes of PPI, including responsive PPI.
PPI is context-specific so it is important to tailor PPI to the emergent needs of trials and be creative to encourage active engagement. Researchers felt that involving contributors beyond an oversight role, that is, not just as a member of the steering committee but in a managerial or responsive capacity helped to foster meaningful PPI. In terms of responsive PPI, liaison with relevant patient panels or groups may be particularly helpful when more diverse perspectives or wider consensus is needed; individuals might also consider whether surveys (eg, of support group members) would be useful in answering 'burning questions', for example, regarding the acceptability of timing or format of interventions or data collection.

*Communication, clarification and interaction*
"I can't understand why they use me. I just sit there bewildered"
▶ Negotiate with contributors at an early stage about what they can bring to the trial and what they want to bring
▶ Determine whether this matches the trial's needs and clarify roles and expectations
▶ Be sensitive to contributors' needs and preferences
Communication between researchers and PPI contributors is crucial at the outset to clarify roles and expectations, and throughout the trial to optimise engagement and provide feedback about contributions. It may be that particular contributors do not have the insights a trial needs, or maybe trialists need to rethink their plans for PPI in the light of experience. Researchers should avoid seeming "dispassionate" during meetings when discussing a particular illness or condition that impacts on the lives of PPI contributors, and make a genuine effort to understand contributors' points of view.

*Language of research*
"Break it down into a language everybody understands."
▶ Minimise and explain jargon;
▶ Provide glossaries and 'translations' where applicable.
Researchers and contributors should discuss their written and verbal communication preferences and how to minimise and explain jargon. Suggestions for minimising jargon included lists of acronyms or glossaries of research terms. PPI contributors should be prepared to speak up if there is a problem and, with the help of researchers, be willing to acquaint themselves with specialist terms over time.

*Budgeting for PPI*
"University didn't want to pay him the money" "We had money in the pot but only for one PPI"
▶ Budget for PPI—think about contributors' time plus expenses.
▶ Explore opportunities for pretrial support for PPI.
Well thought-through plans will help inform how much to 'cost in' for PPI. Consult with administrators in your organisation at an early stage to iron out processes for payments to PPI contributors. Talk to contributors to make sure they will be happy to accept reimbursement beyond expenses. Find out whether there are any local or national resources to support PPI prior to funding applications.

*Fit for purpose PPI*
"The person we chose had very little engagement, it struck me as a complete waste of time"
▶ Agree what type of PPI would be appropriate and understand why.
▶ Consider benefits of involving those with experience of the condition.
▶ Recognise potential drawbacks of involving those under current care of the researcher.
Think through plans for PPI and centre them round the aims and needs of the trial. Agreement about and understanding of *what* and *why* PPI is needed will help in planning it. Involving people with experience of the condition, intervention or service where applicable may be particularly germane in identifying research priorities and enhancing trial design. However, the inclusion of patients under the current care of a team member may lead to difficulties for researchers as well as contributors.

—a related article from our study will fully explore the perceived impact of PPI in this cohort. While oversight PPI seemed limited in terms of its practical impact, arguably it may serve important ethical and moral functions. However, in order to avoid inadvertently promoting PPI that is devoid of any function for researchers and contributors, as we note above, funders should take full account of any PPI which has taken place prior to funding applications as well as encourage applicants to justify future plans for involvement. The NIHR HTA programme states:

"While patient and public involvement (PPI) may not always be needed for all types of research, it is always relevant for HTA trials." http://www.nets.nihr.ac.uk/__data/assets/pdf_file/0003/77160/Preparing-a-full-application-for-the-Clinical-Trials-and-Evaluation-Board.pdf (last accessed 9 March 2014). Even if there is consensus that PPI is relevant for all trials, it may not be relevant at all stages of all trials. Equally, funders may wish to contemplate how 'contingency' resources could be made available for those trials that encounter unexpectedly intense needs for PPI over the course of their implementation.

Our findings add fuel to recent drives and initiatives to promote the assessment and reporting of PPI processes[6][28][30] http://www.journalslibrary.nihr.ac.uk/authors/report-preparation/report-contents/14 including the GRIPP checklist.[41] The CONSORT (Consolidated Standards of Reporting Trials) Statement, which was established specifically to encourage adequate reporting of RCTs, does not cover PPI. We suggest that consideration be given to incorporating advice on reporting of PPI in the main CONSORT checklist, so that reference to PPI is incorporated *within* the main reports of trials, alongside separate detailed reports on PPI, in line with the GRIPP checklist. If, in planning their PPI, trialists are prepared to consider and report its outcomes not only in terms of what happened and how, but also how this matched the needs of the trial, whether any complications arose or adaptations were made, and what lessons were learnt, then the evidence base will grow and the research community as a whole can learn. The EPIC project has highlighted the value of listening to the accounts of PPI contributors as well as researchers, and this should feed into the evaluation and reporting of PPI.

## Conclusions

While most trialists fully implemented their documented plans for PPI there were traces of a minimalist approach. Planning and engaging PPI contributors early, and beyond a primarily oversight role, seems to be the most salient message from this analysis. At the same time some degree of flexibility within plans is prudent, and making allowances for the unexpected may help all stakeholders to make the most of PPI. The involvement of investigators' current patients as PPI contributors should be given cautious consideration as there is the potential for conflict between clinical and research roles. PPI activity prior to funding is as integral to meaningful involvement as PPI activity during trials, and more so in some cases. Proper and flexible planning by research teams will be instrumental in helping them to monitor, adapt and report PPI during and after trials, and in helping the research community as a whole learn how to optimise PPI.

**Author affiliations**
¹Department of Biostatistics, University of Liverpool, Liverpool, UK
²Department of Women's and Children's Health, NIHR Clinical Research Network: Children, Coordinating Centre, University of Liverpool, Institute of Translational Medicine (Child Health), Alder Hey Children's NHS Foundation Trust, Liverpool, UK
³TwoCan Associates, Hassocks, UK
⁴Department of Psychological Sciences, University of Liverpool, Liverpool, UK

**Acknowledgements** The authors acknowledge the help and support of the NIHR HTA in establishing the cohort documentation. Alison Allam, Philip Bell, Heather Goodare and Alison Walker formed the EPIC Patient Advisory Group. All members commented on the interview schedules and manuscript. The authors also acknowledge the late Neil Formstone, former member of the advisory group.

**Collaborators** The EPIC Patient Advisory Group—Alison Allam, Philip Bell, Neil Formstone, Heather Goodare, and Alison Walker.

**Contributors** DB was involved in conducting qualitative data analysis and interpretation and writing the manuscript. CG conceived the idea for the research, led the development of the grant application and project, developed the interview schedules, contributed to interpretation, reviewed drafts of the manuscript, and agreed the final version of the manuscript. LD was involved in developing the interview schedules, recruiting informants to the study, conducting the qualitative interviews, interpreting the findings, reviewing drafts of the manuscript and agreeing the final version of the manuscript. JP contributed to the project specification, development of the interview schedules, interpretation of the findings, commented on the manuscript and led the coordination of the EPIC Patient Advisory Group. BH contributed to the project specification, commented on the manuscript and co-led the coordination of the EPIC Patient Advisory Group. PRW contributed to the project specification and provided comments on the manuscript. BY contributed to the grant application and project specification, designed the qualitative components, led in all aspects of their development, implementation, analysis and interpretation, reviewed all drafts of the manuscript, and agreed the final version of the manuscript.

**Funding** The study was jointly funded by NIHR HS&DR and INVOLVE (project number 10/2001/29). The study was conducted independently of the funders and competing interests have been declared. The University of Liverpool were sponsors of the project.

**Competing interests** BH reports grants from the National Institute for Health Research, during the conduct of the study. BH received personal fees from NIHR, the Medical Research Council, and the User Involvement Shared Learning Group. DB, LD, CG. JP and PW have nothing to disclose.

**Ethics approval** University of Liverpool Institutional Ethics Board.

**Provenance and peer review** Not commissioned; externally peer reviewed.

**Data sharing statement** No additional data are available.

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
