## [Reviewer comments · BMJ Open]

ARTICLE DETAILS

TITLE (PROVISIONAL)	From plans to actions in patient and public involvement: Qualitative study of documented plans and the accounts of researchers and patients sampled from a cohort of clinical trials
AUTHORS	Buck, Deborah; Gamble, Carrol; Dudley, Louise; Preston, Jennifer; Hanley, Bec; Williamson, Paula; Young, Bridget

VERSION 1 - REVIEW

REVIEWER	Dr Jonathan Boote University of Sheffield, UK
REVIEW RETURNED	29-Aug-2014

GENERAL COMMENTS	This is an interesting paper describing a qualitative study which analysed, from a cohort of trials, plans for public involvement and reflections on its implementation from the perspectives of the researcher(s) and the PPI contributor(s). The study reported in this paper is novel and makes a useful contribution to the literature on public involvement. The paper is well written and interesting to read. I have a few suggestions for the authors in terms of how they might improve their paper 1. I think the abstract, if there is sufficient space, should include a section describing the sample of researchers and PPI contributors who participated in the study2. I would prefer the term 'patient involvement' rather than 'patient partnerships' in the opening paragraph of the introduction3. I think, in the introduction, the authors may wish to reconsider or rephrase their statement that there are uncertainties about how PPI could best be implemented in the context of a trial. INVOLVE has produced a guidance document on PPI in trials (ref 17 in the paper) and a recent narrative review has discussed published case examples of PPI in clinical trials http://ijj.cgpublisher.com/product/pub.88/prod . In addition, a recent paper has set out a proposed standard operating procedure for how clinical trials units may wish to incorporate PPI in trials (ref 28 in the paper).4. 2nd paragraph in the introduction, I think the authors should perhaps soften their remarks about PPI causing problems to research validity due to bias. Researchers also have the potential to bring their own biases into their studies, so this isn't just a problem caused by PPI contributors.5. In the methods section, I think the authors should explain why interview transcripts were not returned to participants.6. Page 7, trial steering committee should have the acronym in brackets as the acronym was used later on in the manuscript.7. Page 8, TM should be spelled out in a full at first usage – I assume it stands for trial manager?8. Figure 1 page 9, it is not clear to me how the final figure of 28 trials eligible for analysis was derived. Also it would be useful for the diagram to explain the reasons for the drop out between those agreeing to be interviewed and the number who were actually interviewed.9. In the 'previous research' part of discussion, the authors may wish to reflect on their findings with respect to previous research into the attitudes of researchers to PPI, which suggested there is epistemological dissonance (or a 'know-do gap') between what
--

	researchers understand about PPI and what they actually do about it in practice (see Thompson et al 2010 http://onlinelibrary.wiley.com/doi/10.1111/j.1369-7625.2009.00532.x/abstract;jsessionid=37A716C9C1C06A08DE0DA5BF0BD2C9B9.f01t01?deniedAccessCustomisedMessage=&userIsAuthenticated=false) and Ward et al 2011 http://jos.sagepub.com/content/46/1/63.short 10. page 21, the PiiAF is not just about implementing PPI, but also about evaluating the impact of PPI 11. page 21, the authors' remarks about the need for flexibility and contingency planning in PPI are most interesting, and this to me raises important issues for funders; the implication being: should/could trialists (and other researchers) be allowed to request additional funding for extra PPI when the need becomes apparent during the course of a study. I would encourage the authors to elaborate on this issue in the 'implications for funders' section. 12. In the introduction to Box 1, I would suggest to the authors that they recommend readers to consider their recommendations alongside existing PPI good practice and guidance that have already been published, such as INVOLVE's guidance on PPI and trials, and the SOP for PPI in RCTs mentioned above, and also the consensus-derived principles of successful PPI in health research (Boote et al, 2006 http://www.sciencedirect.com/science/article/pii/S0168851005000837 ; Telford et al, 2004 http://onlinelibrary.wiley.com/doi/10.1111/j.1369-7625.2004.00278.x/abstract?deniedAccessCustomisedMessage=&userIsAuthenticated=false) 13. In box 1, under 'flexible PPI', I found the statement, 'consider whether surveys (e.g. of support group members) would be useful in answering 'burning questions, or qualitative research to gain deeper understanding' to be a little unclear. I think the authors should give specific examples of what 'burning questions' might be in the context of the design and conduct of a trial, and how qualitative research techniques could be used to bolster PPI. The authors need to be very careful how they talk about using research techniques in the context of PPI as this could blur the lines between what is 'research' and what is 'PPI' 14. Page 24, the authors may wish to cite 2 recent papers reporting and reflecting on researchers' use of PPI bursaries from the NIHR Research Design Services (Boote et al, 2013 http://onlinelibrary.wiley.com/doi/10.1111/hex.12130/abstract?deniedAccessCustomisedMessage=&userIsAuthenticated=false ; Walker and Pandya Wood, 2013 http://onlinelibrary.wiley.com/doi/10.1111/hex.12127/abstract) to support their argument that these bursaries exist, but that researchers with tight submission deadlines have difficulties accessing these . 15. Page 24, the EQUATOR network has recently undertaken work to adapt and simplify the GRIPP checklist, to assist researchers in reporting the impact of PPI in all studies, including that of trials. So the authors may wish to revise their recommendation of incorporating PPI within the CONSORT checklist.
--	---

REVIEWER	David Evans University of the West of England, Bristol, UK
	I led another study of public involvement in research funded by the same NIHR programme as the authors.
REVIEW RETURNED	02-Nov-2014

GENERAL COMMENTS	This is a very good paper which adds original data to deepen our understanding of PPI in clinical trials research. There are just a few relatively minor aspects I think the authors need to address before it would be acceptable for publication. First, the authors describe conducting a thematic analysis using the Framework approach in their Methods section, but what these themes are is not detailed in their Results section, nor is the Results section structured around these themes. This seems an important omission and I would like to more explicit reporting of the thematic
---

	analysis and greater clarity on how it informed the structuring of the Results section. Second, the authors report that in nine of the 28 trials they were able to conduct interviews with both the CI and a PPI contributor. It would be helpful to know how consistent the accounts of the CI and PPI contributor were in these cases. In one previous study of PPI in NHS research there was some divergence between researchers and PPI contributors on how the latter were involved in the research (Barber et al 2007 Involving consumers successfully in NHS research: a national survey. Health Expectations 10(4): 380-91). Third, the authors make clear that some of the CIs engaged in tokenistic or tick box approaches to PPI. I would have liked to see a bit more discussion of this and its implications, particularly in the context of recent research into researchers attitudes towards PPI in research (Thompson et al 2009 Health researchers' attitudes towards public involvement in health research. Health Expectations 12(2): 209-220; Ward et al 2009 Critical perspectives on 'consumer involvement in health research. Journal of Sociology 46(1): 63-82). The authors offer helpful tips for planning and implementing PPI in clinical trials (box 1), but these presuppose that CIs want to genuinely engage with PPI; what are the implications for PPI contributors and funders when CIs are only tokenistic?
--	--

VERSION 1 – AUTHOR RESPONSE

Reviewer Name Dr Jonathan Boote

This is an interesting paper describing a qualitative study which analysed, from a cohort of trials, plans for public involvement and reflections on its implementation from the perspectives of the researcher(s) and the PPI contributor(s). The study reported in this paper is novel and makes a useful contribution to the literature on public involvement. The paper is well written and interesting to read.
Response: We thank the reviewer for this positive summary of our paper.

I have a few suggestions for the authors in terms of how they might improve their paper

1. I think the abstract, if there is sufficient space, should include a section describing the sample of researchers and PPI contributors who participated in the study

Response: Thank you, this would have been a useful addition but we had already reached the word limit of 300 words for the abstract.

2. I would prefer the term 'patient involvement' rather than 'patient partnerships' in the opening paragraph of the introduction

Response: We have amended this term as suggested.

3. I think, in the introduction, the authors may wish to reconsider or rephrase their statement that there are uncertainties about how PPI could best be implemented in the context of a trial. INVOLVE has produced a guidance document on PPI in trials (ref 17 in the paper) and a recent narrative review has discussed published case examples of PPI in clinical trials

<http://iji.cgpublisher.com/product/pub.88/prod> . In addition, a recent paper has set out a proposed standard operating procedure for how clinical trials units may wish to incorporate PPI in trials (ref 28 in the paper).

Response: Thank you. We have re-phrased this statement to some extent. While we agree that there is a growing number of resources available, these are based on experiences of particular groups

rather than evidence. The guidance, the SOP, and the narrative review that are mentioned by the reviewer (all of which we now reference) are incredibly helpful resources, but we do still believe that uncertainty remains for many researchers.

4. 2nd paragraph in the introduction, I think the authors should perhaps soften their remarks about PPI causing problems to research validity due to bias. Researchers also have the potential to bring their own biases into their studies, so this isn't just a problem caused by PPI contributors.

Response: This is a fair point and in light of this we have removed the remark about PPI as a source of bias and now refer to the source of such difficulties as arising from how PPI is drawn upon to inform research, rather than necessarily from PPI itself.

5. In the methods section, I think the authors should explain why interview transcripts were not returned to participants.

Response: Returning transcripts to participants is a practice that some researchers believe is important for respondent validation, although like many aspects of qualitative research there is no consensus regarding this. We concur with those who argue that, viewed from a broadly interpretive approach to qualitative research, this practice is not without problems. In particular, the "feedback" generated by this form of respondent validation constitutes additional data that requires further interpretation. We have briefly noted this in the manuscript and added a reference that refers to this issue.

6. Page 7, trial steering committee should have the acronym in brackets as the acronym was used later on in the manuscript.

Response: Thank you, we have rectified this.

7. Page 8, TM should be spelled out in a full at first usage – I assume it stands for trial manager?

Response: We had defined 'TM' earlier in the Methods (Design) section.

8. Figure 1 page 9, it is not clear to me how the final figure of 28 trials eligible for analysis was derived. Also it would be useful for the diagram to explain the reasons for the drop out between those agreeing to be interviewed and the number who were actually interviewed.

Response: We have revised Figure 1 and hope that it is now clear how the final figure of 28 trials eligible for analysis was derived. We have also added information which explains the reasons for the 'drop out' between those agreeing to be interviewed and the number actually interviewed.

9. In the 'previous research' part of discussion, the authors may wish to reflect on their findings with respect to previous research into the attitudes of researchers to PPI, which suggested there is epistemological dissonance (or a 'know-do gap) between what researchers understand about PPI and what they actually do about it in practice (see Thompson et al 2010

[http://onlinelibrary.wiley.com/doi/10.1111/j.1369-](http://onlinelibrary.wiley.com/doi/10.1111/j.1369-7625.2009.00532.x/abstract;jsessionid=37A716C9C1C06A08DE0DA5BF0BD2C9B9.f01t01?deniedAccessCustomisedMessage=&userIsAuthenticated=false)

[7625.2009.00532.x/abstract;jsessionid=37A716C9C1C06A08DE0DA5BF0BD2C9B9.f01t01?deniedAccessCustomisedMessage=&userIsAuthenticated=false](http://onlinelibrary.wiley.com/doi/10.1111/j.1369-7625.2009.00532.x/abstract;jsessionid=37A716C9C1C06A08DE0DA5BF0BD2C9B9.f01t01?deniedAccessCustomisedMessage=&userIsAuthenticated=false)) and Ward et al 2011

<http://jos.sagepub.com/content/46/1/63.short>

Response: Thank you, we now reflect upon this earlier work in our 'Previous research' section.

10. page 21, the PiiAF is not just about implementing PPI, but also about evaluating the impact of PPI

Response: We have added this point, thank you.

11. page 21, the authors' remarks about the need for flexibility and contingency planning in PPI are most interesting, and this to me raises important issues for funders; the implication being:

should/could trialists (and other researchers) be allowed to request additional funding for extra PPI when the need becomes apparent during the course of a study. I would encourage the authors to

elaborate on this issue in the 'implications for funders' section.

Response: This is a very good point, thank you. We have expanded upon this in the 'implications for funders' section, as encouraged.

12. In the introduction to Box 1, I would suggest to the authors that they recommend readers to consider their recommendations alongside existing PPI good practice and guidance that have already been published, such as INVOLVE's guidance on PPI and trials, and the SOP for PPI in RCTs mentioned above, and also the consensus-derived principles of successful PPI in health research (Boote et al, 2006 <http://www.sciencedirect.com/science/article/pii/S0168851005000837> ; Telford et al, 2004 <http://onlinelibrary.wiley.com/doi/10.1111/j.1369-7625.2004.00278.x/abstract?deniedAccessCustomisedMessage=&userIsAuthenticated=false>

Response: Another fair and important comment which we have taken on board in revising our manuscript.

13. In box 1, under 'flexible PPI', I found the statement, 'consider whether surveys (e.g. of support group members) would be useful in answering 'burning questions, or qualitative research to gain deeper understanding' to be a little unclear. I think the authors should give specific examples of what 'burning questions' might be in the context of the design and conduct of a trial, and how qualitative research techniques could be used to bolster PPI. The authors need to be very careful how they talk about using research techniques in the context of PPI as this could blur the lines between what is 'research' and what is 'PPI'

Response: We have provided an example of what a burning question might be in the context of trial conduct or design. After further consideration we have taken out the reference to qualitative research as we agree there is much contention and potential for confusion surrounding the respective functions and contributions of qualitative research and PPI in the context of health research. While this is an important issue, addressing it would be beyond the scope of our paper.

14. Page 24, the authors may wish to cite 2 recent papers reporting and reflecting on researchers' use of PPI bursaries from the NIHR Research Design Services (Boote et al, 2013 <http://onlinelibrary.wiley.com/doi/10.1111/hex.12130/abstract?deniedAccessCustomisedMessage=&userIsAuthenticated=false> Walker and Pandya Wood, 2013 <http://onlinelibrary.wiley.com/doi/10.1111/hex.12127/abstract>) to support their argument that these bursaries exist, but that researchers with tight submission deadlines have difficulties accessing these.

Response: We are very grateful to have been made aware of these 2 important articles which we now cite, as recommended.

15. Page 24, the EQUATOR network has recently undertaken work to adapt and simplify the GRIPP checklist, to assist researchers in reporting the impact of PPI in all studies, including that of trials. So the authors may wish to revise their recommendation of incorporating PPI within the CONSORT checklist.

Response: Thank you for this suggestion. While we appreciate that the GRIPP guidelines are being adapted, these do not yet appear to be published and are not currently prominent on the EQUATOR website. Therefore we find it difficult to consider revising our recommendation. We feel it is valid to highlight the potential value of future revisions of CONSORT which may incorporate recommendations about the reporting of PPI in the main trial reports (not just in separate reports). Moreover, many journals require a completed CONSORT checklist on submission of an article if the study is a RCT, and of course CONSORT is specific to trials, as was our study. Also, in the longer term, it would be desirable to see a level of consistency between various guidelines.

Reviewer Name David Evans

Institution and Country University of the West of England, Bristol, UK

Please state any competing interests or state 'None declared': I led another study of public involvement in research funded by the same NIHR programme as the authors.

This is a very good paper which adds original data to deepen our understanding of PPI in clinical trials research. There are just a few relatively minor aspects I think the authors need to address before it would be acceptable for publication.

Response: We thank the reviewer for these encouraging comments on our paper.

First, the authors describe conducting a thematic analysis using the Framework approach in their Methods section, but what these themes are is not detailed in their Results section, nor is the Results section structured around these themes. This seems an important omission and I would like to more explicit reporting of the thematic analysis and greater clarity on how it informed the structuring of the Results section.

Response: The Results section does not directly list the categories or themes that we generated in the course of our analysis as we did not feel this was the most appropriate approach for the present paper. Our primary aim was to investigate how officially documented plans for PPI (e.g. the type and timing of PPI) compared with informants' accounts of what subsequently happened. Also, while we asked informants about their experiences and views of PPI, we did not directly ask them to compare what was planned with what happened, as this may have given rise to idealised accounts. Thus, our case-by-case comparison and analysis focussed on patterns in what PPI was implemented compared to what was planned. We feel the important categories in describing these patterns, and in informing the practice of trialists and contributors, were the type of PPI, the challenges encountered in implementing PPI and the lessons learnt/suggestions. The presentation of our findings did not lend itself neatly to a presentation organised around a list of themes from the qualitative interviews and we feel such a presentation would detract from the clarity of our paper. We believe that the illustrative quotes in the main text and the extensive information provided in Table 1 allow the reader to judge the integrity of our analysis. However, to avoid confusing readers who expect the mention of "thematic analysis" to necessarily lead to a presentation of findings structured around a list of themes, we have amended the description of the analysis in the methods.

Second, the authors report that in nine of the 28 trials they were able to conduct interviews with both the CI and a PPI contributor. It would be helpful to know how consistent the accounts of the CI and PPI contributor were in these cases. In one previous study of PPI in NHS research there was some divergence between researchers and PPI contributors on how the latter were involved in the research (Barber et al 2007 Involving consumers successfully in NHS research: a national survey. *Health Expectations* 10(4): 380-91).

Response: We agree that the consistency or otherwise between CI and PPI contributor accounts is of much interest. We did find good concordance, which we now mention in the first paragraph of the Results, but further exploration of this was beyond the remit of the current paper. We are also mindful that, while some trials may have had more than one PPI contributor, only one was usually interviewed by us, and that memory played a part in the recollections of both researchers and contributors. A 'sister paper' from our study which is under review elsewhere, and which will describe perceptions of the impact of PPI, will cover 'triangulation' between researcher and contributor accounts in more detail.

Third, the authors make clear that some of the CIs engaged in tokenistic or tick box approaches to PPI. I would have liked to see a bit more discussion of this and its implications, particularly in the context of recent research into researchers attitudes towards PPI in research (Thompson et al 2009 Health researchers' attitudes towards public involvement in health research. *Health Expectations*

12(2): 209-220; Ward et al 2009 Critical perspectives on 'consumer involvement in health research. Journal of Sociology 46(1): 63-82).

Response: Thank you, we acknowledge that we have not fully discussed the 'tick box' approaches that some of the researchers talked about. This was intentional as we were aware that it would have been too repetitive of the sister paper that we refer to above. This 'impact paper' will go into more detail and discussion about tokenism. We realise that the reviewer would not be aware of this fact, and we thank him for acknowledging how important this issue is and for drawing our attention to the 2 key references, (i.e. Thompson et al 2009 and Ward et al 2009), which we now cite in our 'Previous research' section.

The authors offer helpful tips for planning and implementing PPI in clinical trials (box 1), but these presuppose that CIs want to genuinely engage with PPI; what are the implications for PPI contributors and funders when CIs are only tokenistic?

Response: Thank you for this very valid point, which we have now acknowledged in relation to our point about the potential usefulness of the material in Box 1 for informing judgements about PPI at the funding stage.

* * * * *

Finally, we have made a small number of additional minor changes to the wording of our manuscript in places where we felt these would improve clarity. All of these are indicated in track changes.

Once again, we would like to thank the editor and reviewers for their time and thoughtful comments which we hope we have adequately addressed.